# A Vision-Based Fuzzy Control to Adjust Compression Speed for a Semi-Dieless Bellows-Forming

**Sugeng Supriadi [1],\*, Tsuyoshi Furushima [2] and Ken-ichi Manabe [3]**

[1] Department of Mechanical Engineering, Universitas Indonesia, Jakarta 16424, Indonesia
[2] Department Mechanical and Bio-functional Systems, Tokyo University, Tokyo 153-8505, Japan; tsuyoful@iis.u-tokyo.ac.jp
[3] Department of Mechanical Systems Engineering, Tokyo Metropolitan University, Tokyo 192-0397, Japan; manabe@tmu.ac.jp
\* Correspondence: sugeng@eng.ui.ac.id

**Abstract:** A novel semi-dieless bellows forming process with a local heating technique and axial compression has been initiated for the past years. However, this technique requires a high difficulty in maintaining the output quality due to its sensitivity to the processing conditions. The product quality mainly depends on not only the temperature distribution in the radial and axial direction but also the compression ratio during the semi-dieless bellows process. A finite element model has clarified that a variety of temperature produced by unstable heating or cooling will promote an unstable bellows formation. An adjustment to the compression speed is adequate to compensate for the effect of the variety of temperatures in the bellows formation. Therefore, it is necessary to apply a real-time process for this process to obtain accurate and precise bellows. In this paper, we are proposing a vision-based fuzzy control to control bellows formation. Since semi-dieless bellows forming is an unsteady and complex deformation process, the application of image processing technology is suitable for sensing the process because of the possible wide analysis area afforded by applying the multi-sectional measuring. A vision sensing algorithm is developed to monitor the bellows height from the captured images. An adaptive fuzzy has been verified to control bellows formation from 5 mm stainless steel tube in to bellows profile up to 7 mm bellows height, processing speed up to 0.66 mm/s. The adaptive fuzzy control system is capable of appropriately adjusting the compression speed by evaluating the bellows formation progress. Appropriate compression speed paths guide bellows formation following deformation references. The results show that the bellows shape accuracy between target and experiment increase become 99.5% under given processing ranges.

**Keywords:** bellows forming; vision-based sensor; fuzzy control; semi-dieless forming; local heating

## 1. Introduction

Metal bellows are convoluted metal tube that provide high flexibility on various direction. It is widely applied in the flexible joining of piping systems for water, oil and gas provisions. Metal bellows are usually produced through a hydroforming or a gas-forming process. When we increase the internal pressure of the tube, the bellows shape will be formed following the die shape. A new process of manufacturing expansion joint bellows from Ti-6Al-4V alloys using a gas pressure [1]. A fluid pressure instead of a gas pressure in the metal semi-dieless bellows forming process [2]. A single-step tube hydroforming process for producing metal bellows has been conducted with specific dies to fabricate a rectangular, circular, and triangular bellows profile [3]. However, not only do those methods require expensive dies and a complex machine but they also are inefficient in manufacturing a small quantity or various sizes of bellows. I the previous work, a novel technique had been proposed to fabricate

metal bellows without employing internal fluid pressure or dies. The new technique only employed a compression-assisted local heating using a high-frequency induction heating source [4]. An observation on deformation mechanism of semi-dieless bellows forming using finite element analysis has been verified that bellows formation was caused by localized heating on top of the convolution due to the effect of various heating techniques that could be employed using induction heating [5,6]. Since the deformation is not guided using dies, the forming phenomena is very vulnerable to the processing conditions such as temperature, compression ratio, and cooling process. As a result, the produced bellows have low dimensional/shape accuracies of convolutions. Low efficiency is one of the technical issues in this process. Therefore, in this study, a vision-based fuzzy control system for semi-dieless bellows forming was developed to improve bellows accuracy and repeatability. Finite element analysis was also observed to establish appropriate control for this process.

## 2. The Principle of Semi-Dieless Bellows Forming through Local Heating

Semi-dieless bellows forming with a local heating technique is a novel process of producing a bellows shape without the uses of internal pressure and dies. This process is borrowed from previous technology of dieless drawing process where using force and localized heat to produce incrementally metal deformation such as stainless steel, magnesium and ceramic [7–12]. However, the forming principle of semi-dieless belows forming is entirely different from all other applied loading and deformation phenomena. Figure 1 shows the schematic of semi-dieless bellows forming with a local heating technique [5]. A combination of local heating and compression moving promotes continuous local bucking deformation of a tubular workpiece with a mandrel inner tube. We start the process of the semi-dieless bellows forming by thickening the tube under a heating coil. Afterwards, the local buckling is induced and moved to the cooling region as the first convolution. Simultaneously, a new local buckling occurs behind the first convolution leading to the second convolution. This process is repeated until a series of convolutions produces bellows. The local buckling results from the low flow stress of the material at an elevated temperature. We obtain the compressive load by applying the compression speed $v_1$ faster than the feeding speed $v_2$ ($v_1 > v_2$). To measure the convolution formation progress, the compression ratio (C) is used, as in $C = v_1/v_2$. After the first convolution formed by the buckling passes through the cooling area, the next convolution will be initiated behind the first convolution located inside the heating coil. When the convolutions are continuously produced in the tube, bellows are produced. We can control the bellows height by selecting the compression ratio [4,5]. The increasing compression ratio increases the bellows height to a specific value, while the bellows pitch remains constant. The mechanism has been applied for non-ferrous material such aluminum with similar behavior [13]. Effect when using bigger tube size has been verified that able to produce bellows using semi dieless bellows forming with similar characteristic [14].

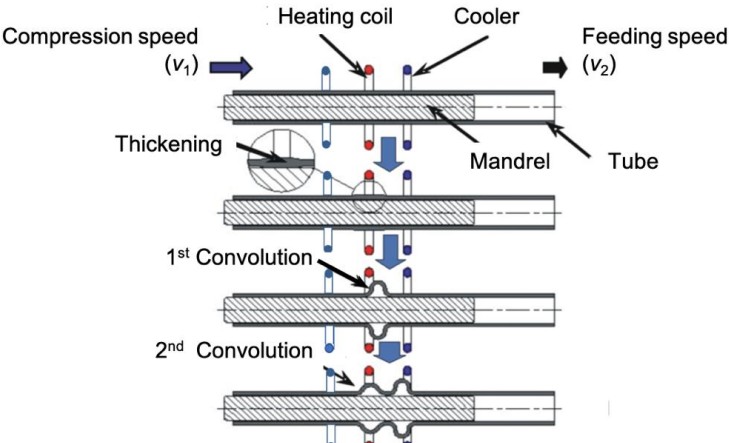

**Figure 1.** Schematic of the semi-dieless bellows forming with a local heating technique.

Deformation mechanism of the semi-dieless forming process is under a free deformation. Therefore, this technique is very sensitive to the forming conditions and disturbances. As a result, the bellows height and pitch are not constant and do not vary as shown in Figure 2a,b. Another problem that occurs is asymmetric bellows as shown in Figure 2c. These phenomena also happened in aluminum tube [13].

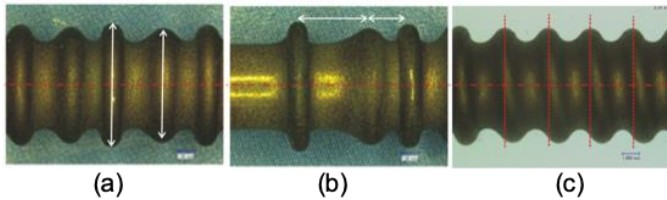

**Figure 2.** The problem of semi-dieless bellows forming on SUS 304 (**a**) Variety of bellows height, (**b**) Variety of bellows pitch, and (**c**) Asymmetric bellows shape.

## 3. Finite Element Analysis (FEA)

To evaluate the dieless bellows forming process and to verify problems in the dieless bellows forming process, we have developed the finite element model to simulate the dieless bellows forming process by using a finite element method. Figure 3 shows the schematic of an axisymmetric bellows forming model on the FEM carried out on an MSC Marc Mentat version 2005 commercial software from MSC Software USA. A deformable tube and a rigid mandrel are comprised of 5-node, asymmetrically quadrilateral shell elements. We simulate local heating by applying heat flux on the tube surface. The magnitude of heat flux in the induction heating is inversely proportional to the distance from the coil to the workpiece surface. Therefore, Equation (1) is utilized to model various heating when taking into consideration the bellows formation progress already verified in our previous work [5], where $q$ is heating quantity, $a$ is a constant, and $y$ is the position of a node in the y-axis in a global position. $qs$ and $a$ are set to achieve the processing temperature up to 1200 °C. Just before the convolution occurs, the heating quantity is $qs$; after convolution is formed, the heat flux quantity increases. We model a double cooling system by using film subroutines to reduce the temperature by increasing the heat transfer capacity. Figure 3 shows the geometry of heating and cooling system model referring to the experimental conditions with a 3.5-mm cooling distance ($Cd$), a 5-mm cooling length ($Cl$), and a 4-mm heating length ($Hl$).

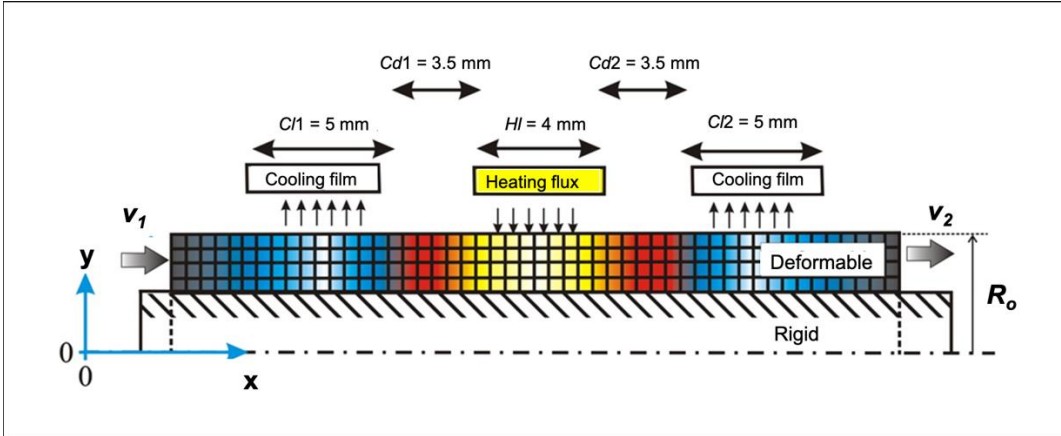

**Figure 3.** Schematic of the finite element model for the asymmetrical semi-dieless bellows forming process.

Taking into consideration strain hardening ($n$), strain rate sensitivity ($m$), and strength coefficient ($K$), we utilize a tube material made of SUS 304 Stainless steel. It was clarified that $n$, $m$, and $K$ (Mpa)

are function constants of temperature as shown in Equations (2)–(5), where, $\sigma$ is flow stress (Mpa), $\varepsilon$ is strain, and $\dot{\varepsilon}$ is strain rate [15]. We assume that the material is isotropic in mechanical and thermal properties. $v_1$ (t) is set at 0.6 mm/s, while $v_2$ (t) is set constantly at 0.3 mm/s. While boundary condition for the FEA is shown in Table 1.

$$q = q_0 + (a \cdot (4-y)^2) \tag{1}$$

$$\sigma = K \cdot \varepsilon^n \cdot \dot{\varepsilon}^m \tag{2}$$

$$K(T) = (7.10^{-4}\cdot T^2) - (2.14\cdot T) + 1788.3 \tag{3}$$

$$n(T) = (-2.472\cdot10^{10}\cdot T^3) + (4.645\cdot10^2\cdot T^2) + 0.4367 \tag{4}$$

$$m(T) = (4.374\cdot T) + 1.449\cdot10^{-3} \tag{5}$$

**Table 1.** Material model and Boundary conditions of SUS 304 used in Finite Element Model (FEM).

| Boundary Conditions | |
|---|---|
| Heating quantity, $q$ (W·mm$^{-2}$) | Equation (1) |
| Heating temperature (°C) | 1100 |
| Heating length, $Hl$ (mm) | 5 |
| Cooling Length, $Cl$ (mm) | 5 |
| Heat transfer coefficient of cooling, $h_c$ (W·mm$^{-2}$·K$^{-1}$) | 1000 |
| Heat transfer coefficient of radiation to air, $h_a$ (W·mm$^{-2}$·K$^{-1}$) | 30 |
| Thermal conductivity, $\lambda$ (W·mm$^{-1}$·K$^{-1}$) | 0.0163 |
| Specific heat (J·g$^{-1}$·K$^{-1}$) | 0.502 |
| Mass density,$\rho$ (g·mm$^{-3}$) | 0.008 |

We promote deformation in the semi-dieless bellows forming by adding the compression speed and localized elevated temperature. By assuming that the compression speed driven by the servo motor is stable, we suspect that the cause of deformation instability is variations in the processing temperature. In the real semi-dieless bellows forming process, heating system and cooling system effect to temperature stability. The unstable heating system results from a variety of distances between the heating coil and the workpiece. On the other hand, the unstable cooling is produced by the unstable water cooling. To verify these effects on the semi-dieless forming process, we model a variety of temperatures by varying the quantity of heat flux ($q_{sv}$ (W/mm$^2$)) and heat transfer in the cooling area ($h_{cv}$ (W·mm$^{-2}$·K$^{-1}$)) as shown in Equations (6) and (7) respectively. Figure 4 shows the schematic of variety of heating and cooling in time series.

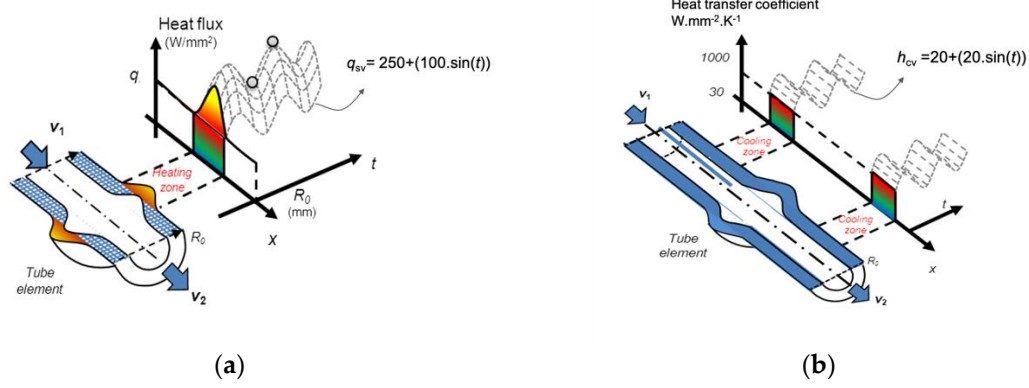

**(a)**                                   **(b)**

**Figure 4.** (**a**) Schematic of a variety of heating ($q_s$) and (**b**) a variety of cooling system ($h_c$).

$$q_{sv} = 250 + (100\cdot\sin t) \tag{6}$$

$$h_{cv} = 20 + (20 \cdot \sin t) \tag{7}$$

Various heating and cooling are modeled using a sinusoidal function as shown in Equations (6) and (7), while the visualization of the heating and cooling models in time series is shown in Figure 5.

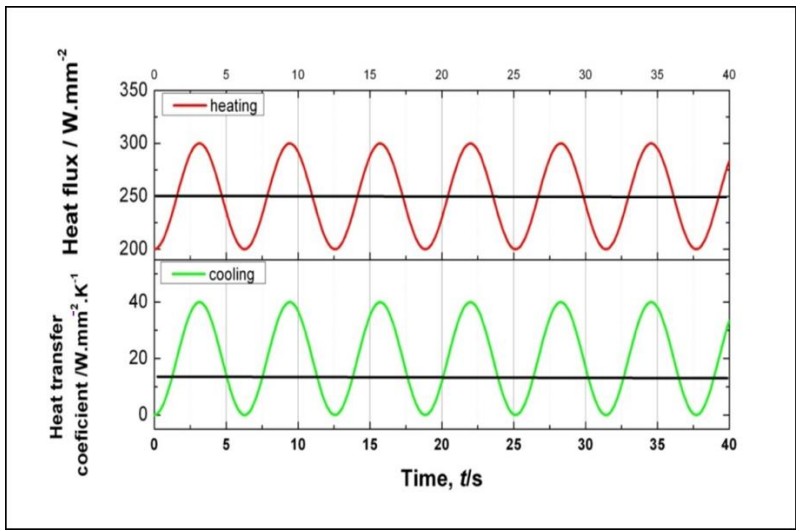

**Figure 5.** Various models of heating and cooling in time series.

The results of the simulation using the variation of heating and cooling condition produce different bellows profile as shown in Figure 6. Changes in temperature distributions promoted by a variety of heating and cooling affect the bellows profile stability of the bellows pitch and height. We obtain the bellows height by measuring the outer diameter of the bellows profiles. Compared to the constant heating and cooling, various heating and cooling produce various bellows heights and pitches. When a temperature difference occurs in the radial direction, an asymmetric bellows shape is created.

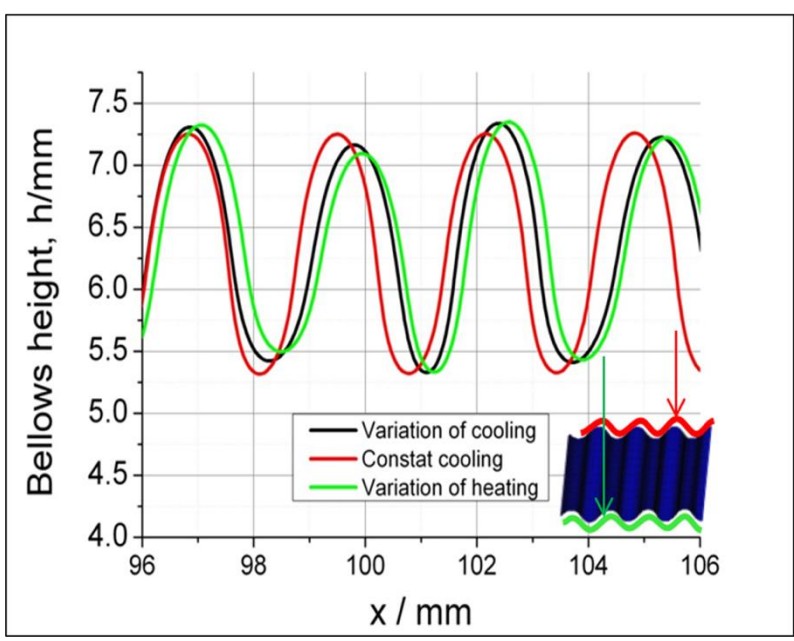

**Figure 6.** Bellows profiles under different conditions.

## 4. Selecting the Compensation Method for the Unstable Deformation Promoted by a Variety of Temperatures

We have clarified the unstable deformation resulting from a variety of temperatures by using a Finite element model (FEM). We have also conducted a FEM to find appropriate compensations for the unstable deformation. We model this unstable deformation by varying the temperatures obtained from various heat transfers. We propose two compensation methods of adjusting the heating flux and adjusting the compression ratio to compensate for those various temperatures. Since the feeding speed ($v_2$) affects the processing speed and stability, $v_2$ is kept constant. Therefore, we adjust the compression ratio by adjusting the compression speed ($v_1$).

The results of the two-compensation methods using the heating power and compression speed are shown in Figure 7. The results indicate that adjustment to the heating power cannot achieve the desired effect, as shown in Figure 8 since not only does the bellows temperature depend on the heating and heating condition, but it also depends on the bellows height. We promote the deformation of the semi-dieless bellows forming by localizing the heating on top of the convolution while applying a compression stroke. Therefore, when compensating the temperatures compensated from the adjustment to the heat flux, we have difficulties in adjusting to the bellows height due to the fact that the temperature during the process also performs a non-linear delay. Therefore, it is difficult to implement and regulate the temperature; it will require further studies on temperature behaviors.

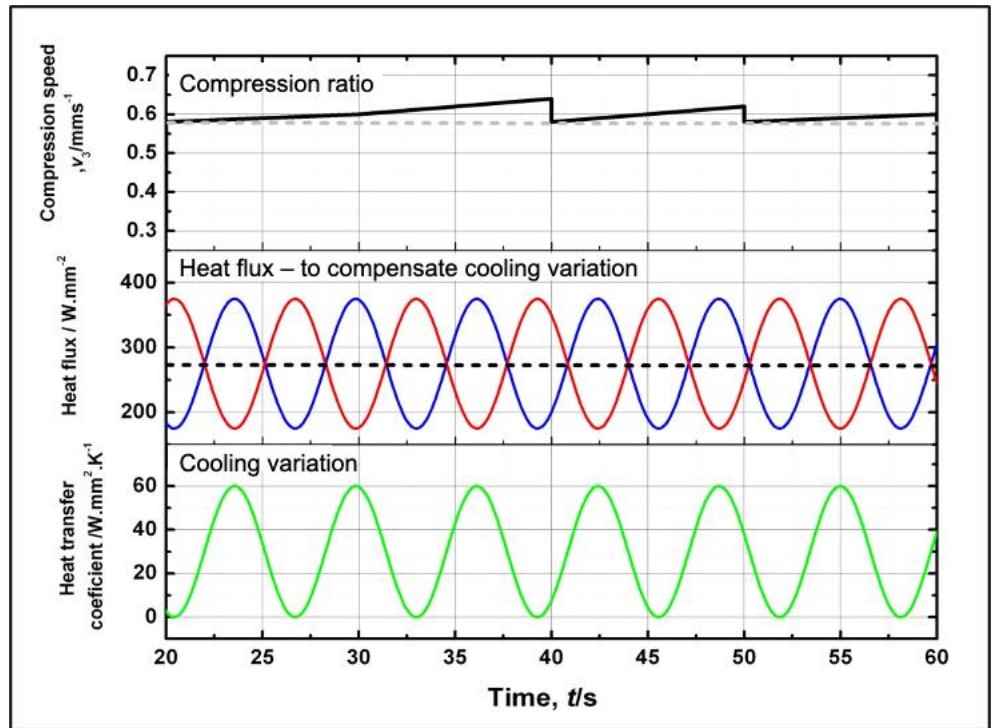

**Figure 7.** Compensations under various, different temperatures through applying changes in the heat flux and the compression speed.

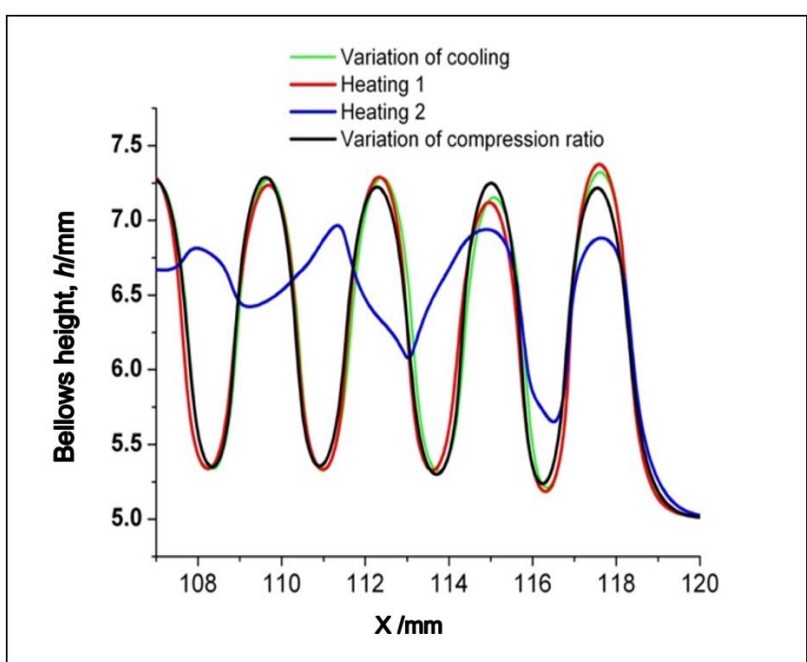

**Figure 8.** Results of the heating and the compensation of the compression speed to various cooling.

On the other hand, adjustment to the compression speed can effectively and efficiently modify the bellows height. When we use the compression speed to adjust the bellows height, we can obtain a homogeneous bellows height. Figure 9 shows the effects of the compression stroke (since the compression speed is higher than the feeding speed, it significantly affects the bellows height). Therefore, we should adjust the bellows height by adjusting compression stroke ($v_1$ or $v_2$).

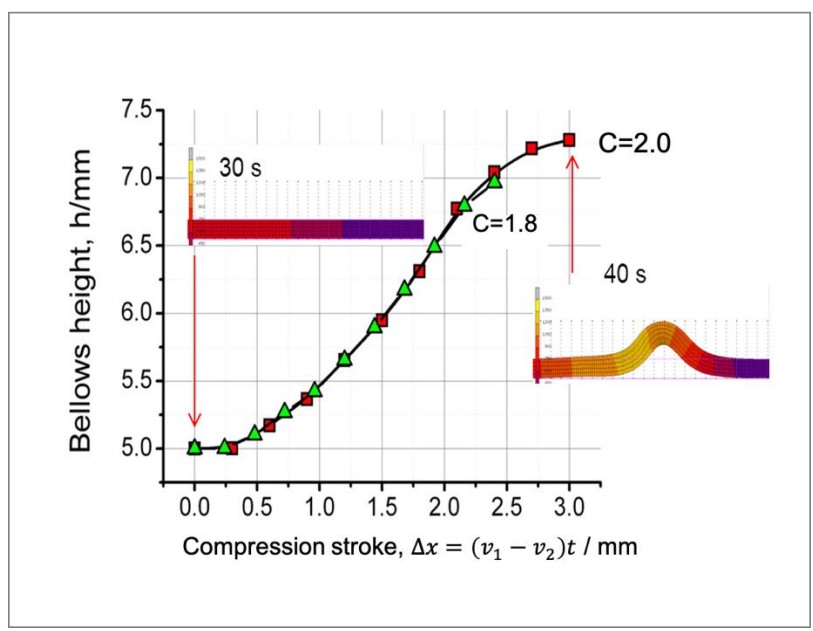

**Figure 9.** Effects of the compression stroke on the bellows heights.

## 5. Improving the Dieless Bellows Forming Process with the Application of a Feedback Control System

Since the deformation of semi-dieless bellows forming is a free and complex deformation, a feedback control system is a reasonable choice to improve the process and eliminate problems

during the process. We have to establish components for the feedback control system such as the deformation reference, the sensing system, and the controller as shown in Figure 11. In the previous work, we had developed a feedback control system for a dieless drawing process by using an adaptive fuzzy logic control [16]. However, in this work, the difference lies in deciding on the deformation target of the bellows profile and selecting the fuzzy control parameters to obtain the best results. Since the deformation characteristic of semi-dieless bellows forming is different from that of the previous work, we have to decide on another control strategy.

### 5.1. Deformation Reference

In the real bellows geometry, the outer diameter of the bellows target shape is a function of the $x$ domain ($D(x)$). In the real-time monitoring, the control program is run under a time series. Therefore, it is necessary to construct a diameter reference in the time domain. According to the sequence of bellows formation that we had obtained by employing a FEM from our previous work, bellows formation is a cyclic process. A series of bellows height has a specific period as shown in Figure 10.

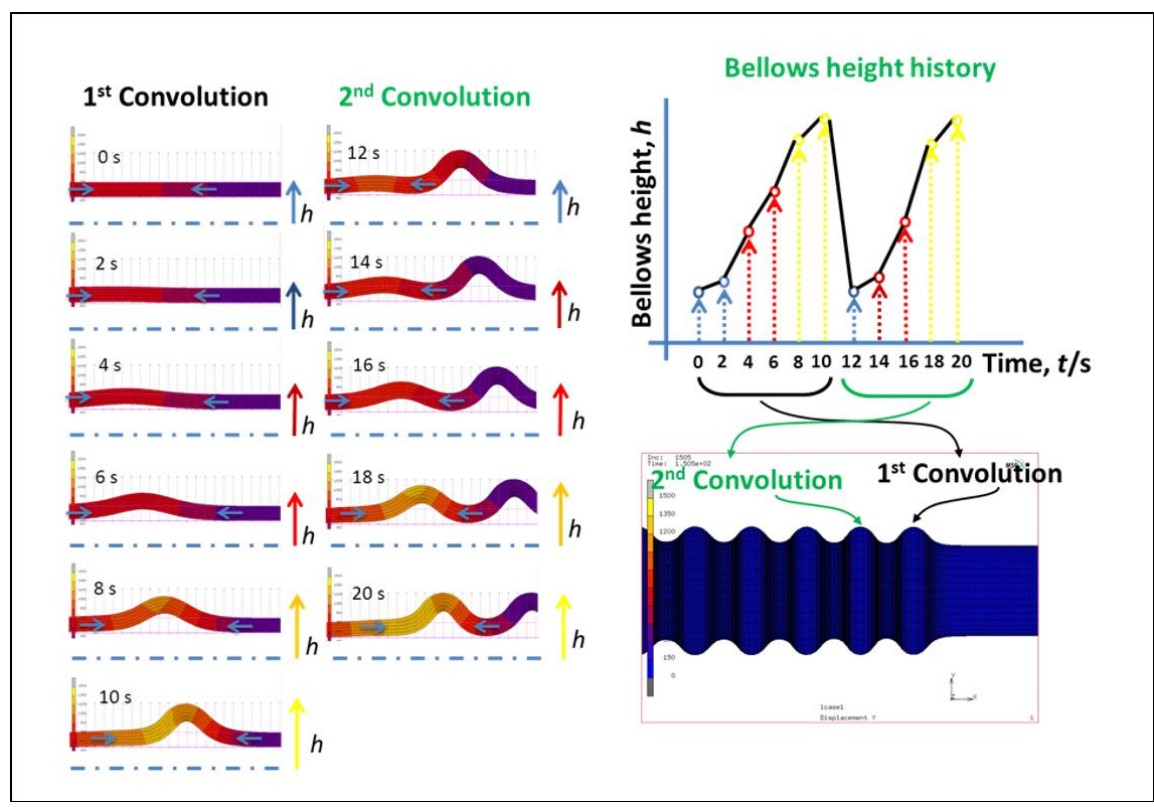

**Figure 10.** FEM results showing bellows formation history.

In this paper, the reference of bellows height progress uses a sinusoidal function. The bellows height reference in a time series is expressed in Equation (8), where $D$ is the bellows height, *Amp* is the amplitude of bellows, $\omega$ is the frequency of bellows, $\phi$ is the phase shift, $D_0$ is the initial diameter, and $t$ is time (s).

$$D(t) = Amp \sin(\omega t - \phi) + D_0 \tag{8}$$

### 5.2. A Vision-Based Feedback Control

A Continuous Semi-dieless bellows forming has complex mechanism. Therefore, deformation is free deformation due to elevated temperature and compression force. The deformation area of Continuous semi dieless bellows forming is wide and move. Therefore, it requires a contactless measurement technique to monitor deformation progress in continuous semi-dieless bellow forming.

An imaging-based sensing system appropriate sensing technique that compromise measuring condition in continuous semi-dieless bellows forming [17]. An image is captured with an infrared filter to improve focused on the object at elevated temperature [18,19]. An infrared-pass filter removes visible light passing through the image sensor that improves detection on the edge of bellows. An Algorithm has been adopted from dieless drawing process. Therefore, semi dieless bellows forming need additional rule to measure dynamic bellows height formation [20].

The components in the vision-based feedback control are bellows image capturing, image and data processing, calibration of the machine vision. Since difficulties in capturing deformation zone inside heating coil, the camera is tilted 10° from the perpendicular direction. In this study, the picture is captured using an infrared-pass filter. Therefore, a high contrast image of the workpiece by can be obtained for image and data processing.

### 5.3. Establishing a Control System

Fuzzy inference is an artificial intelligent computation method in which linguistic calculation is converted into a mathematical calculation. Fuzzy sets improve the handling of uncertainty by reducing it and developing a correct conclusion although the real world is not precise or specific. The fuzzy inference is selected since this method is suitable for controlling a non-linear system with limited knowledge of the system. The fuzzy control system provides transparency, and the intuitive nature of the rule base and input variables adopted to make it relatively easy to be developed, tested and modified [21]. To establish the appropriate fuzzy control system of the semi-dieless bellows forming, a conventional fuzzy and adaptive fuzzy logic control is prepared. The conventional fuzzy has fixed parameters during the control process such as membership function and fuzzy rules while the adaptive fuzzy logic control has adjustable parameters such as membership function and fuzzy rules according to the deformation characteristics (bellows progress, error and changing error).

Figure 11 shows the control flowchart of two fuzzy control strategies for semi-dieless bellows forming. The fuzzy input is the diameter error ($e$), changing rate of error ($\Delta\dot{e}$), and bellows formation progress in percent ($h$) only for adaptive fuzzy control. The outputs of fuzzy control are the compression speed. The adaptive fuzzy logic control has a supervisory system to adjust the membership span of main fuzzy control by using gain value. The gain is used to modify the membership span of compression speed. The membership functions for error, changing rate of error, $\Delta v_1$, and gain are shown in Figure 12. Membership of each category has specific name that used for making fuzzy rule in the controller. Each member ship consist of range of parameter (x axis) relative to level of membership value (from 0 to 1). The membership value 0 means the parameter is not affect to fuzzy inference calculation. The membership value depends on parameter (x axis) of membership function. The rule logic represents an expert of dieless bellows forming. The Fuzzy rule is construct based on previous works and FEA. There are two fuzzy controller that applied in this paper, a conventional fuzzy controller, and an adaptive fuzzy controller.

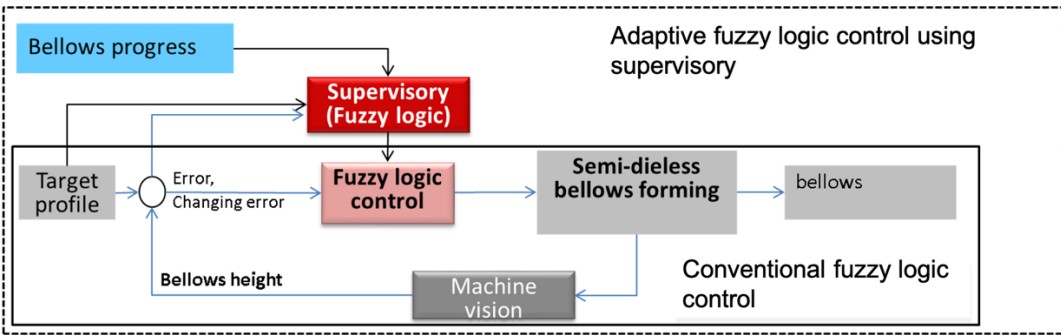

**Figure 11.** Flowchart of the semi-dieless bellows forming with a vision-based fuzzy control.

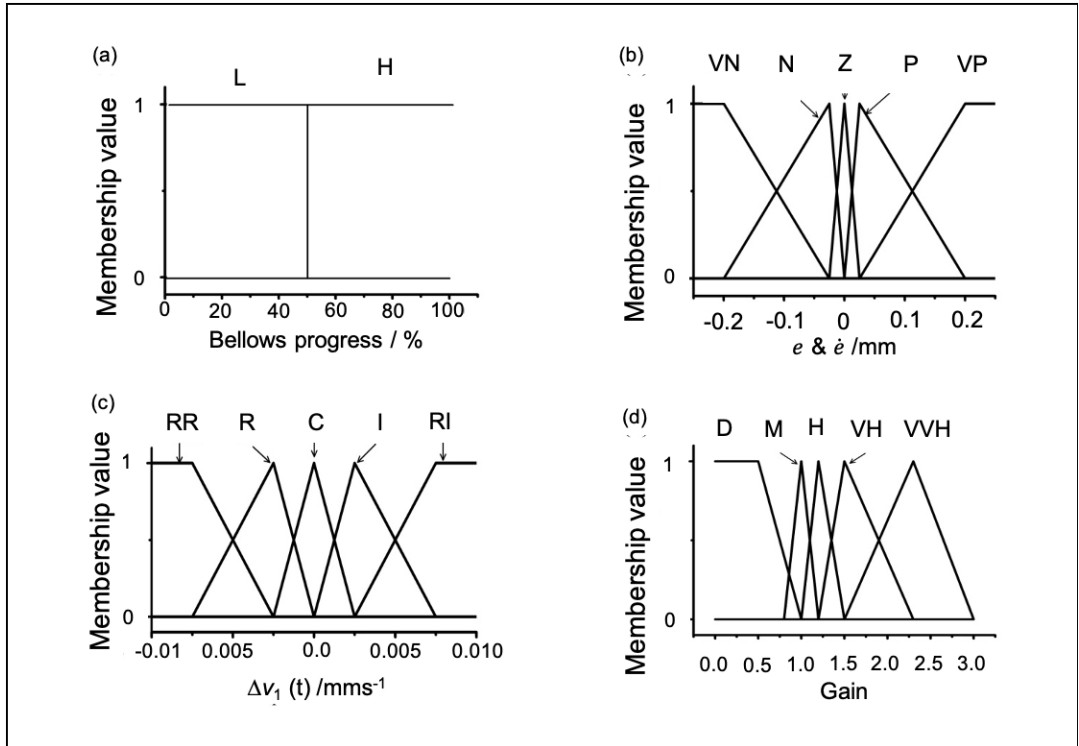

**Figure 12.** Membership functions of conventional fuzzy controller: (**a**) bellows formation progress consist of two membership function L (Low), H (High), (**b**) bellows height error and changing rate of the error has similar membership function VN (Very Negative), N ( Negative), Z (Zero), P (Positive), VP (Very Positive), (**c**) incremental compression speed ($\Delta v_1$) consist of membership function RR (Rapid Reduce ), R (Reduce), C (constant), I (Increase), RI (Rapid Increase), and (**d**) gain consist of membership function D (Decrease), M (Medium), H (High), VH (Very High), and VVH (Very Very High).

The conventional fuzzy rule is used to adjust compression speed according to the error and changing error value as shown in Table 2. When machine vision measure the belows height, then compare with target profile resulting error ($e_h$) and changing error ($\Delta e_h$). As the input, value of $e_h$ and $\Delta e_h$ will determine membership function in Figure 12b. According to fuzzy rule in Table 2, membership function of $e_h$ and $\Delta e_h$ determines membership function of output on incremental compression speed ($\Delta v_1$). The fuzzy inference calculates $\Delta v_1$ to adjust compression speed ($v_1$). Therefore compression speed will be adjust according to error and changing error of bellows simuntanously until bellows formation finish. Membership function on the adaptive fuzzy rules are divided into two different parts, below and over 50% of the bellows height. This strategy is adopted because, in the early stage of bellows formation, the error tends to be positive owing to the process characteristic in bellows formation. Thus, the fuzzy rules are not appropriate when the positive error of $v_,$ is reduced which makes the convolution height low since the deformation area moves. In the case of the control action command for below 50% of the convolution progress, the compression speed, $v_1$, is set as a slow response to the error. If convolution progress is over 50%, the fuzzy rules will be configured as a fast response to the error.

**Table 2.** Conventional fuzzy rules to determine the compression speed adjustment $\Delta v_1$. (*Refer to Figure 12).

| Input | | Output | Input 2: Changing Error ($\Delta e_h$) | | | | |
|---|---|---|---|---|---|---|---|
| | | | VN* | N* | Z* | P* | VP* |
| Input 1: Bellows height error ($e_h$) | VP* | $\Delta v_1$ | I | I | R | R | RR |
| | P* | $\Delta v_1$ | I | C | C | R | R |
| | Z* | $\Delta v_1$ | I | C | C | R | R |
| | N* | $\Delta v_1$ | I | C | C | C | I |
| | VN* | $\Delta v_1$ | RI | I | R | I | I |

Tables 3 and 4 show the details of adaptive fuzzy rules to adjust the $v_1$ and gain in accordance with the fuzzy input of the convolution progress, the diameter error, and the changing rate error. $v_2$ is kept constant since it affects the reference target. The fuzzy rule is divided into two conditions: below and above 50% of the reduction progress. Fuzzy rules for below 50% are set as the slow response because, in this step, the error is always positive. Therefore, if the fuzzy response is high, $v_1$ will decrease. Consequently, it is difficult to achieve a maximum bellows height.

**Table 3.** Adaptive fuzzy rules under 50% of the bellows progress to determine the compression speed adjustment $\Delta v_1$ and gain. (*Refer to Figure 12).

| Input | | Output | Input 2: Changing Error ($\Delta e_h$) | | | | | | | | | |
|---|---|---|---|---|---|---|---|---|---|---|---|---|
| | | | VN* | | N* | | Z* | | P* | | VP* | |
| Input 1: Bellows height error ($e_h$) | VP* | $\Delta v_1$/gain | RI | /H | I | /M | I | /M | C | /M | R | /H |
| | P* | $\Delta v_1$/gain | RI | /M | C | /M | C | /M | C | /M | C | /M |
| | Z* | $\Delta v_1$/gain | C | /M | C | /M | C | /M | C | /M | R | /M |
| | N* | $\Delta v_1$/gain | C | /M | C | /M | C | /M | I | /M | RI | /H |
| | VN* | $\Delta v_1$/gain | R | /H | C | /M | I | /M | I | /M | RI | /VH |

**Table 4.** Adaptive fuzzy rules over 50% of the bellows progress to determine the compression speed adjustment $\Delta v_1$ and gain. (*Refer to Figure 12).

| Input | | Output | Input 2:Changing Error ($\Delta e_h$) | | | | | | | | | |
|---|---|---|---|---|---|---|---|---|---|---|---|---|
| | | | VN* | | N* | | Z* | | P* | | VP* | |
| Input 1: Bellows height error ($e_h$) | VP* | $\Delta v_1$/gain | I | /H | I | /M | R | /M | R | /H | RR | /H |
| | P* | $\Delta v_1$/gain | I | /H | C | /M | C | /M | R | /M | R | /H |
| | Z* | $\Delta v_1$/gain | I | /M | C | /M | C | /M | R | /M | R | /H |
| | N* | $\Delta v_1$/gain | I | /M | C | /M | C | /M | C | /M | I | /M |
| | VN* | $\Delta v_1$/gain | RI | /H | I | /M | R | /M | I | /M | I | /H |

## 6. Experimental Method

### 6.1. Experimental Equipment

The Vision based fuzzy controller is implemented in the horizontal semi-dieless compression machine, a computer for implementing the fuzzy rule, control speed of two servo motors. The machine is equipped with an induction heating source, as shown in Figure 13. The compression speed $v_1$ and feeding speed $v_2$ of the specimen is controlled by rotation of ball screw from servo motor. A narrow heating area is achieved with a high-frequency induction heating apparatus using a 4-mm induction coil width ($Hl$). A water-cooling system is applied to maintain a constant temperature distribution with a 7.5-mm cooling distance ($Cd$). The image acquisition system using a commercial CCD camera to grab the image up to 30 fps. The image of bellows progress is transferred to image processing unit. The fuzzy algorithm is designed to adapt with bellows formation progress. Image processing unit and

fuzzy logic controller is construct under platform of LabVIEW software. The output obtained by the fuzzy inference is an input to the servo motors to adjust the compression speed of $v_2$.

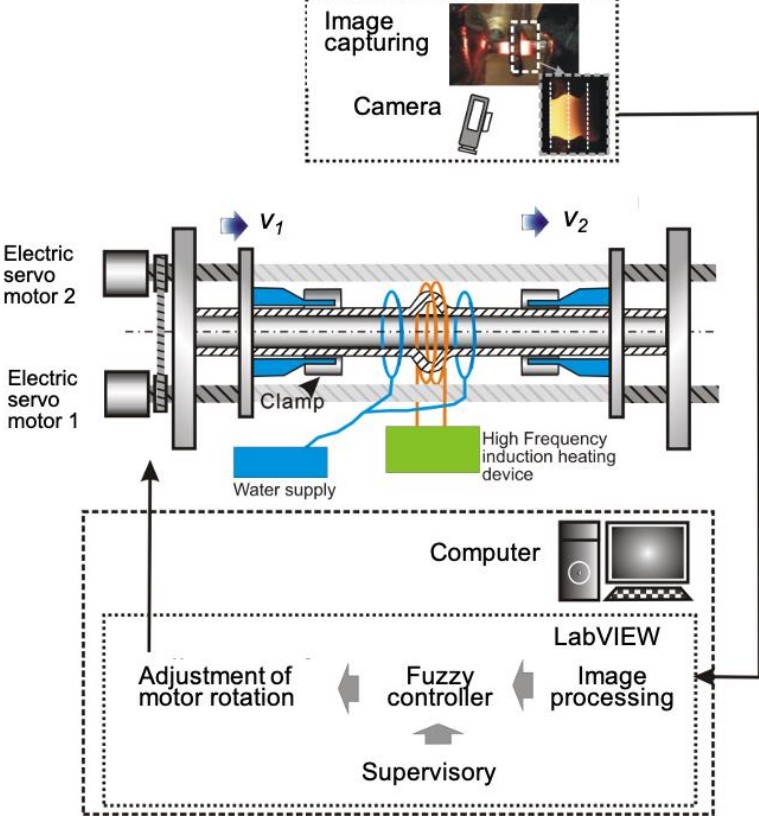

**Figure 13.** Experimental apparatus for the fuzzy control using a machine vision system.

### 6.2. Materials and Experimental Conditions

A stainless-steel, SUS304 tube with a 5-mm outer diameter of and a 0.5-mm-thick wall is used in this experiment. The maximum processing temperature is 1100 °C. We use a mandrel with a 3.9-mm outer diameter to ensure a stable axisymmetric deformation of the convolutions by inserting it into the workpiece. The feeding speed is kept constant at 0.3 mm/s.

In the case of the bellows target shape shown in Figure 14, the bellows shape parameters in the x domain must be converted to that of the time domain where the original diameter, $D_0$ is 5 mm and the maximum bellows diameter is 7 mm. After transformed into Equation (8), the amplitude, $A$, will amount to 1 mm. Pursuant to a previous study, the pitch distance between the convolutions on these processing conditions are fixed at 2.7 mm [4,5].

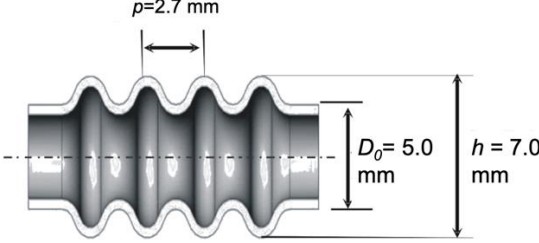

**Figure 14.** Geometry and dimensions of the metal bellows target with the example of $h$ = 7.0 mm.

$$t_{\text{pitch}} = p/v_2 \qquad (9)$$

$$\omega = 2\pi/t_{\text{pitch}} \tag{10}$$

With the 0.3-mm/s feeding speed ($v_2$), the time required to make one convolution is 9.0 s ($t_{\text{pitch}}$). However, for the target reference, only half of the sinusoidal function (the positive slope) is used; it means that a half reference equals one convolution. Therefore, the period of one unit wave is twice as long as the time required to make one convolution. The period of the target reference in the time domain is 18.0 s. Therefore, the reference frequency is 0.348 obtained from Equations (9) and (10). Since the function is sinusoidal and the convolution starts from the lowest value, the function should be shifted 90° to the right ($\phi = 1.57$ rad). The diameter reference converted to the time domain for the given bellows geometry is shown as Equation (11):

$$D\,(t) = \sin\,(0.348t - \phi) + 5. \tag{11}$$

The reference value is set, where the convolution starts from the outer diameter of the tube and evolves with the increasing axial compression. After the diameter reaches the maximum value, the convolution is held owing to the moving heating zone, a new convolution is initiated, and the phenomenon is repeated. When the derivative of the diameter reference is negative ($dD/dt < 0$), the function will be reset. Figure 15 shows the graph of diameter reference converted to the time domain for the profile target in Figure 9. As a result, the metal bellows are measured using a KEYENCE 900 optical microscope to obtain a bellows profile dimension.

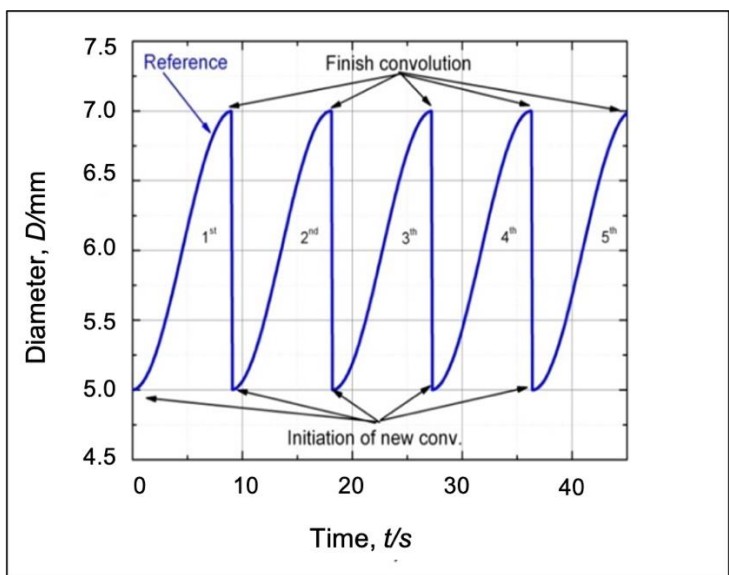

**Figure 15.** Diameter reference used in the semi-dieless bellows forming with vision-based fuzzy control.

## 7. Results and Discussion

Figures 16 and 17 show history of the compression speed paths, the bellows error, the adjustment compression speed ($\Delta v_1$) and the bellows height obtained from the conventional fuzzy and the adaptive fuzzy control respectively. The speed path using a conventional fuzzy is obtained from the adjustment of the initial compression speed according to the error and the changing error condition following the fuzzy rules. This adjustment does not take into considerations the bellows formation progress. The fuzzy control adjusts the compression speed by only taking into considerations the bellows height error from the reference and the actual bellows height. Adjustment of the compression speed follows the fuzzy rules as shown in Table 1. The history of the bellows height produced by using a conventional fuzzy control shows little agreement with the deformation reference and little agreement with the bellows height. It results from the deformation delay in the semi-dieless bellows forming process. The adjustment results of the compression speed in the beginning of the bellows formation will affect

the middle-end bellows formation. In the beginning of the bellows formation, the error is always positive. Therefore, the fuzzy reduces the compression speed. The decreased compression speed reduces the compression stroke that decreases the bellows growth and that approaches the deformation reference. When the history of the bellows height is lower than that in the deformation reference, the fuzzy increases the compression speed. However, due to the delay process and the limited bellows formation timing (10 s), the increased compression speed is not capable of achieving the desired bellows height target before it moves to the cooling zone.

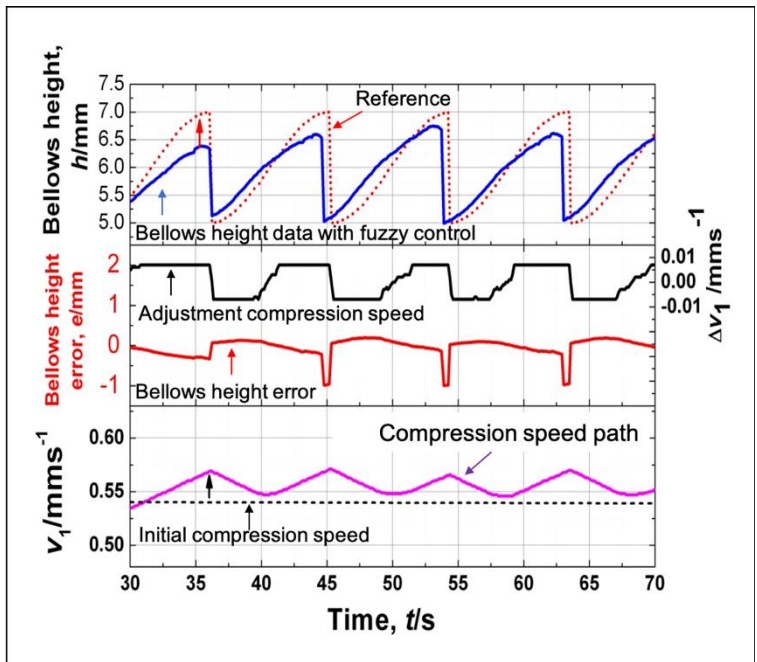

**Figure 16.** Bellows profile produced using a conventional fuzzy control.

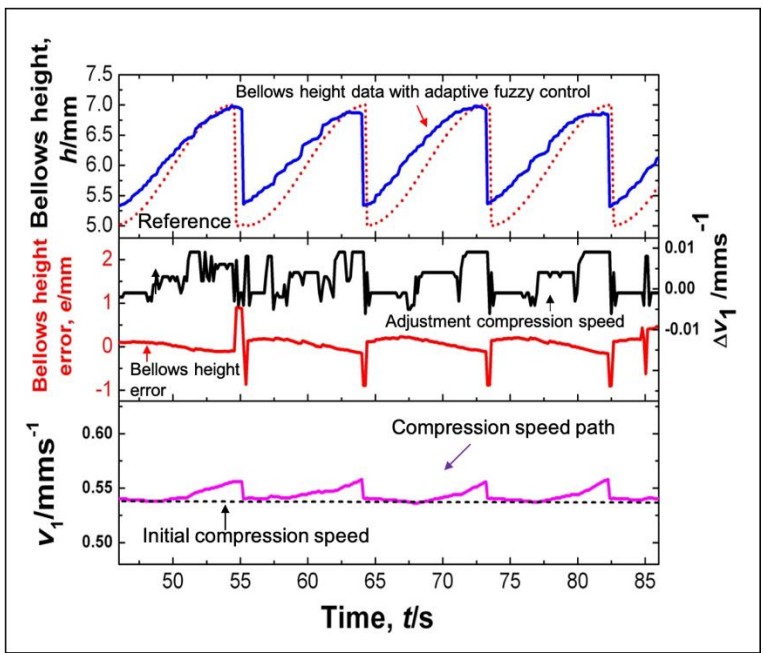

**Figure 17.** Bellows profile produced using an adaptive fuzzy control.

The speed path from the adaptive fuzzy control adjustment fluctuates according to the error and changing error during the semi-dieless bellows forming process. For every convolution formation, however, the compression speed is slightly adjusted when the bellows progress is over 50%, and the compression speed is significantly changed. We apply this strategy to cope with the deformation delay and the positive error in the beginning of each bellows formation. History of the bellows height shows that homogeny and accuracy of the bellows height produced by the semi-dieless bellows forming increases if we apply an adaptive fuzzy control. It shows that from evaluating the bellows progress, applying appropriate adjustment using an adaptive fuzzy control is adequate to control the deformation of the semi dieless bellows forming process.

The vision-based with adaptive fuzzy control performs good agreement with the bellows height target with low variations as shown in Figure 18. It is shows that the compensation based on progress of bellows formation is effective on adjusting bellows progress with low standard deviation. According to FE analysis, bellows height formation can be adjusted effectively after bellows initiation is produce. Adjustment of the compression speed is effective when it is applied to over half of the bellows progress as it has been proven in the FEA. To verify the reliability of the vision-based adaptive fuzzy control system, several experiments have been conducted with different processing conditions such as various feeding speeds, various bellows height targets, and various initial compression ratios.

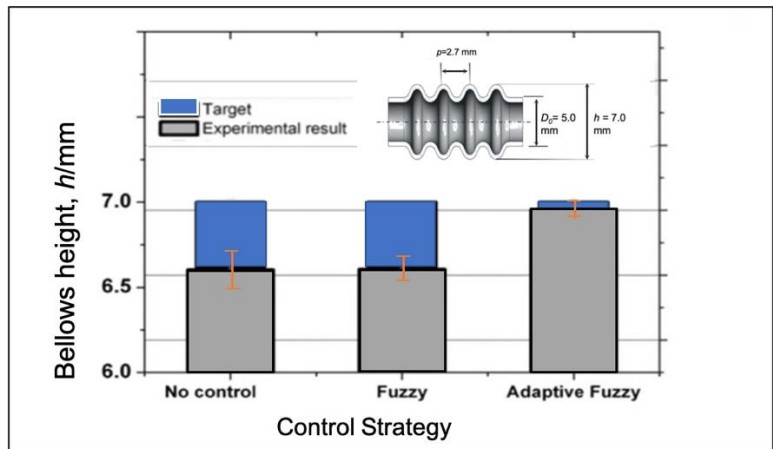

**Figure 18.** Comparison between the bellows height accuracy and its variation under different control strategies.

### 7.1. Verification of the Proposed Control System for the Semi-Dieless Bellows Forming under Various Feeding Speeds

An increased feeding speed reduces the convolution cycle time since the bellows speed formation increases when the feeding speed increases. One cycle of the bellows formation at the feeding speeds of 0.2, 0.3, and 0.5 mm/s are 12.5, 10, and 5 s respectively. Therefore, the time required to produce bellows becomes shorter. The conventional approach produces a lower bellows height, while the adaptive fuzzy controls can adapt to a decreased bellows formation time. Under a given range of the feeding speeds, the proposed control system can maintain the accuracy and homogeny in accordance with the deformation references. The real bellows geometry is analyzed using an optical microscope as shown in Figure 19. The results show that an accurate bellows profile is produced under various feeding speeds at the given range. It indicates that reproducibility of the proposed control system has been verified under different feeding speeds if we perform a similar bellows height and pitch.

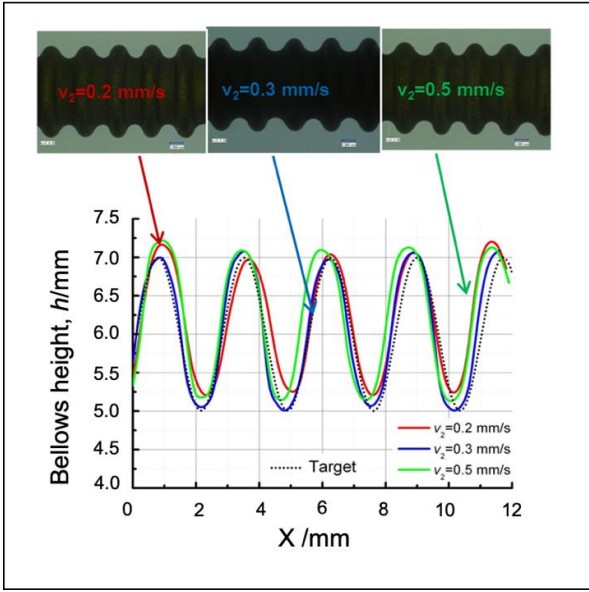

**Figure 19.** Bellows profile at various feeding speeds.

Performance of vision-based adaptive fuzzy control under different initial feeding speed can be seen in Figure 20. All conditions correspond to the target of bellows height. For low speeds, the bellows height tends to be slightly lower than the target height. For high initial compression speeds, the bellows tend to be more elevated than the target even if we apply the adaptive fuzzy to adjust delta $v_1$ in accordance with the error and changing error. Since the adaptive fuzzy is only adjusted for each bellows formation, when one bellows formation is finished, it will be back to the initial feeding speed. Therefore, the initial compression speed is essential and can be changed using a fuzzy. If the bellows height is below the target, the fuzzy will increase the initial feeding speed and the other way around. Figure 21 shows the evaluation of bellows pitch at different initial compression speeds. Even though it produces low accuracy, the proposed control system delivers lower variations compared to that of the conventional approach due to the fact that each bellows formation is a guide that makes every bellows formation more homogeneous.

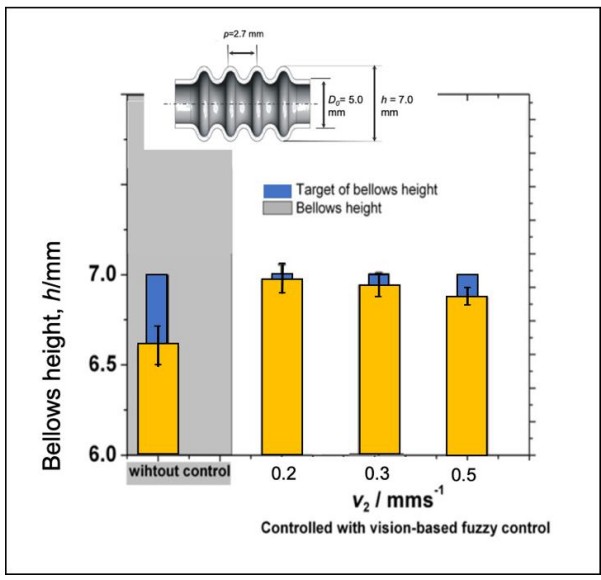

**Figure 20.** Evaluation of the bellows height under different initial compression speeds.

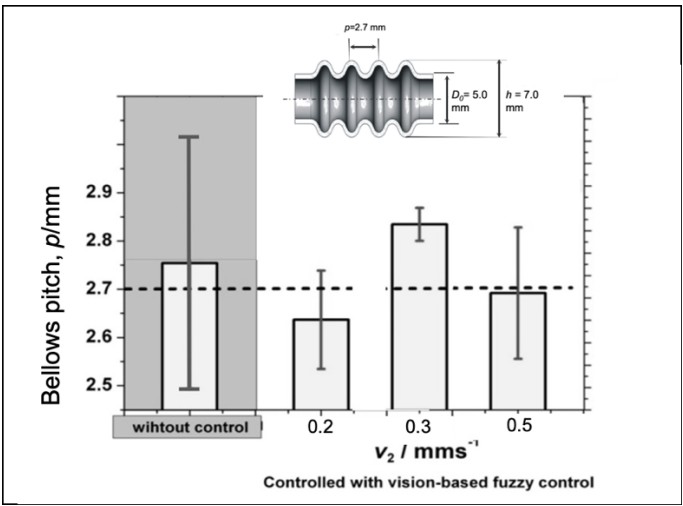

**Figure 21.** Evaluation of the bellows pitches under different initial compression speeds.

### 7.2. Verification of the Proposed Control System for the Semi-Dieless Bellows Forming under Various Bellows Targets

Three different bellows targets have been prepared to verify the performance of the proposed control system. The objectives are indicated with the maximum bellows height (*h*) of the 6.0, 6.5, and 7-mm references. Figure 22 shows that the bellows height data of different targets well correspond to the target. It indicates that the adaptive fuzzy controller with a machine vision sensor has a good performance to enhance the accuracy of the semi-dieless bellows forming. At lower bellows heights, the target shows low accuracy in the beginning of the bellows formation. When we verify this condition with the real bellows profile by using the optical microscope, the bellows show low accuracy in the minimum bellows height and pitch. This behavior results from a thickening process occurring at a low compression ratio as verified in the previous findings. These results indicate that the bellows pitch is not constant [4].

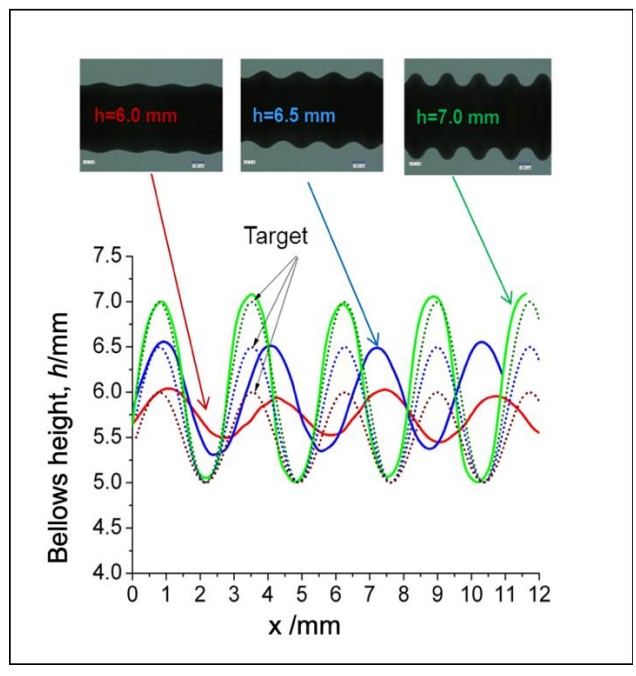

**Figure 22.** Bellows profile at various targets.

Figure 23 shows the comparison between the proposed control system and the conventional approach to control the bellows height. The proposed control system shows an excellent performance to produce a desired bellows height, while, under a similar condition, the conventional approach using a given chart delivers low accuracy and a higher bellows variation. It indicates that the vision based fuzzy control is sufficient to control the bellows height in accordance with the target. However, the proposed control system has a poor performance to achieve the accurate bellows pitch as indicated in Figure 24. Figure 24 shows that the bellows pitch is not constant at various bellows heights as assumed in the previous work.

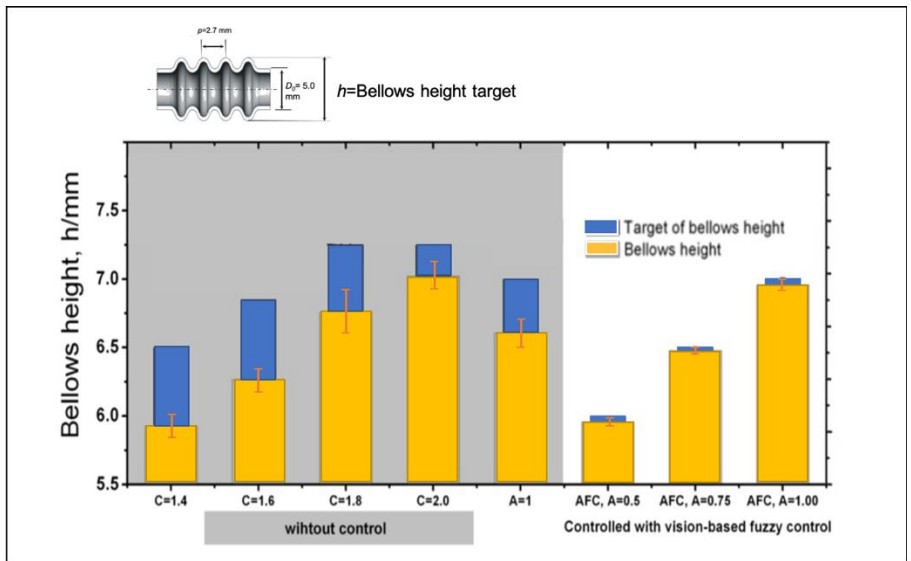

**Figure 23.** Evaluation of the bellows height under different bellows targets.

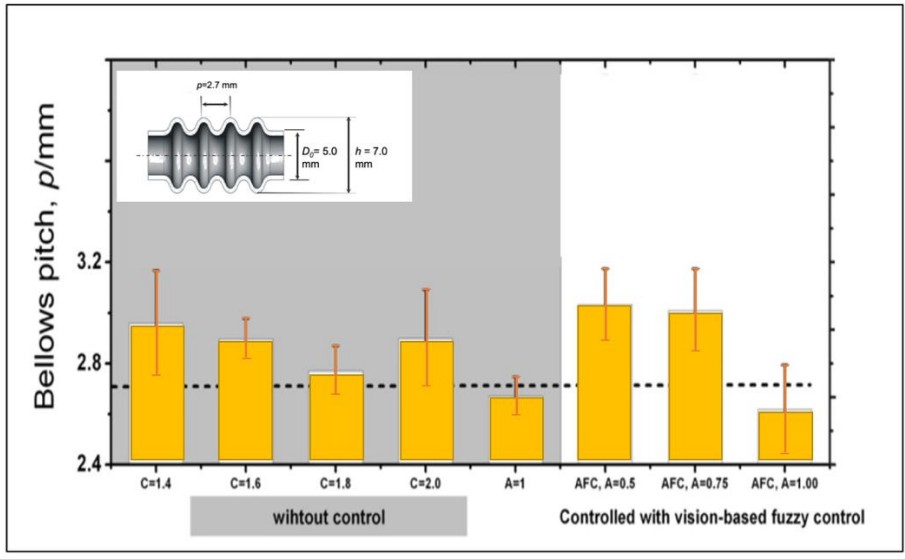

**Figure 24.** Evaluation of the bellows pitches under different bellows targets.

*7.3. Verification of the Proposed Control System for the Semi-Dieless Bellows Forming under Various Initial Compression Speeds*

The proposed control system adjusts the compression speed based on the initial value of the compression speed. Therefore, it is necessary to verify the implementation of the proposed control system when we apply it for different initial set ups of the compression speed or the compression ratio. At lower speeds, the initial compression speed tends to be lower than the bellows height; conversely,

it tends to be higher at a higher initial compression speed. We have verified these findings using an optical microscope to observe the real bellows profile as shown in Figure 25. Not only does a higher initial compression speed perform a higher bellows height but it also performs a shorter bellows pitch. This condition indicates that the fuzzy supervisor should consider an initial compression speed to obtain the appropriate compensation value.

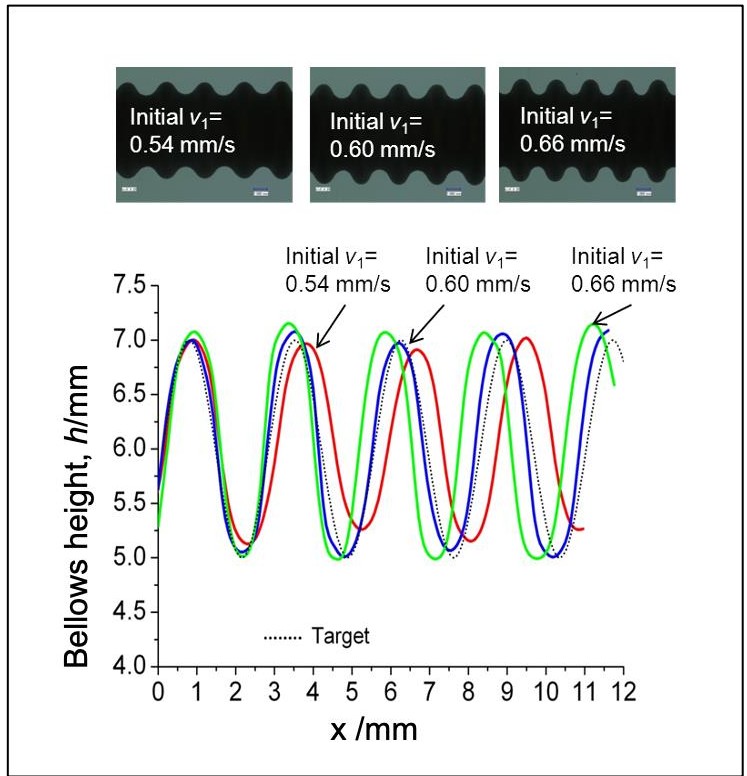

**Figure 25.** Bellows profile at different initial compression ratios.

Making the evaluation on the larger area, we find out that the accuracy of the bellows height will slightly decrease when the processing speed increases as shown in Figure 26 probably since the time to produce one convolution (5 s) is short, so it makes the effectiveness of the vision-based fuzzy control decrease. We can enhance the effectiveness of the adaptive fuzzy by considering the processing speed to adjust the fuzzy parameters such as modifying a membership function span and the adaptive fuzzy to rule at a higher processing speed. Figure 27 shows the evaluation of the bellows pitch that shows different accuracies. However, for a variety of the bellows pitch, the utilization of the proposed control system has produced a lower variation that increases reproducibility.

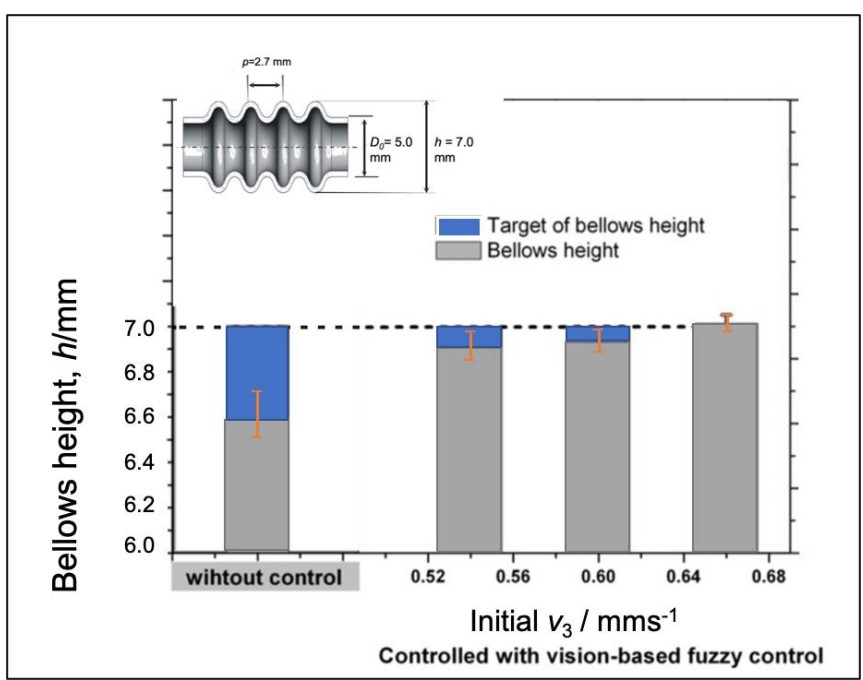

**Figure 26.** Evaluation of the bellows height under different feeding speeds.

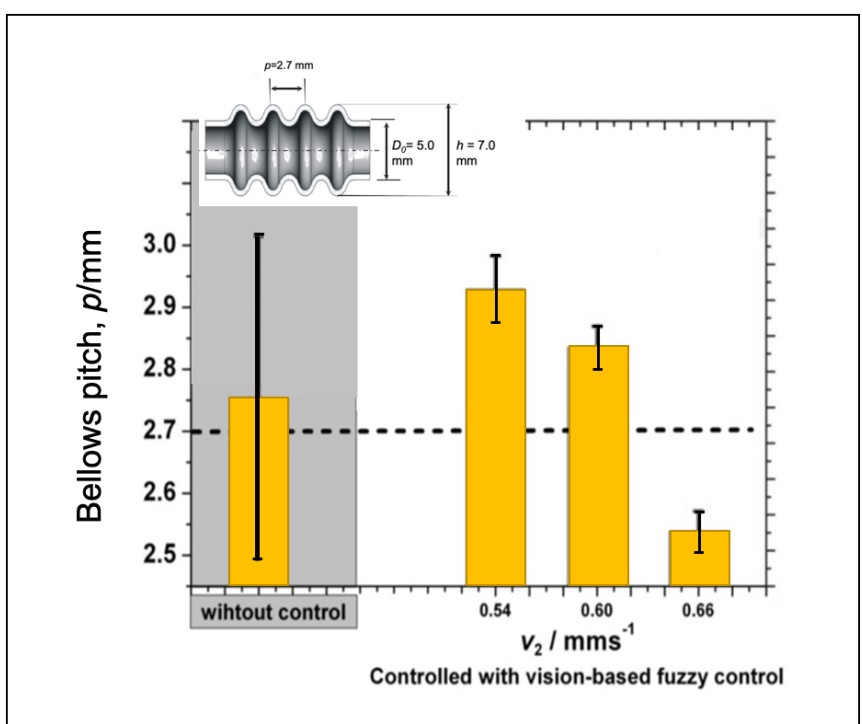

**Figure 27.** Evaluation of the bellows pitch under different feeding speeds.

## 8. Conclusions

We have conducted an investigation to low accuracy of the bellows produced by the semi-dieless bellows forming process by using a finite element method. The results show that low accuracy and low reproducibility of the bellows results from temperature disturbances from an unstable heating or cooling system. Adjusting the compression speed is effective and efficient to compensate for a variety of the bellows during the process created by those temperature disturbances. To implement a real-time adjustment, it is necessary to establish a feedback control system. A vision-based fuzzy control for the

semi-dieless bellows forming has been developed. It is already confirmed that a machine vision is appropriate for the real-time monitoring of the bellows height and pitch. The camera of the machine vision can flexibly be placed and monitor the bellows formation process inside the heating coil to provide deformation data although other measurement devices cannot be applied. The validity of the machine vision has been verified with an optical microscope. The reference target shape for this bellows forming can be used as a sinusoidal function similar to that of the bellows height formation. The adaptive fuzzy controller shows an excellent performance to guide the deformation and to produce accurate bellows. The performance of the proposed feedback control system using vision sensing and an adaptive fuzzy control to provide accurate bellows have been verified under various processing conditions and bellows targets. A remaining problem that still occurs is low accuracy in the bellows pitch due to an incorrect assumption. Further research might be attended to develop a fuzzy supervisor that will consider the initial set up of the dieless bellows forming process.

**Author Contributions:** Supervision, T.F.; Writing–original draft, S.S.; Writing–review & editing, K.-i.M. All authors have read and agreed to the published version of the manuscript.

**Funding:** The APC was funded by Universitas Indonesia: HIbah QQ.

**Acknowledgments:** The author would like to thank to grant from Universitas Indonesia to support this work and Publication.

**Conflicts of Interest:** The authors declare no conflict of interest.

## Nomenclature

| | |
|---|---|
| $A_0$ | : Initial cross section area of tube, $mm^2$ |
| *Amp* | : Amplitude, mm |
| *a* | : Constanta for increasing heat quantity, $W \cdot mm^{-2}$ |
| *Cd* | : Cooling distance, mm |
| *Cl* | : Cooling zone, mm |
| *c* | : Heat capacity, $J \cdot Kg^{-1} \cdot K^{-1}$ |
| *D* | : Diameter, mm |
| $D_0$ | : Initial diameter, mm |
| $e_h$ | : Bellows height error, mm |
| $\Delta e_h$ | : Increment of bellows height error, mm |
| $\bar{e}_h$ | : Average bellows height error, mm |
| G | : Gain |
| h | : Bellows height |
| $h_a$ | : Heat transfer coefficient of radiation to air, $W.mm^{-2} \cdot K^{-1}$ |
| $h_c$ | : Heat transfer coefficient of cooling, $W \cdot mm^{-2} \cdot K^{-1}$ |
| Hl | : Heating length, mm |
| K | : Strength coefficient, MPa |
| *n* | : Strain hardening index |
| *m* | : Strain rate sensitivity |
| $h_{Rev}$ | : Reference bellows height, mm |
| $h_{Prog}$ | : Progress of bellows height, % |
| *q* | : Heat flux quantity, $W \cdot mm^{-2}$ |
| $q_0$ | : Initial heat flux quantity, $W \cdot mm^{-2}$ |
| *p* | : Pitch, mm |
| *t* | : Time, s |
| *X* | : Elongation, mm |
| $t_{pitch}$ | : Time to produce one bellows, s |
| $v_1$ | : Compression speed, $mm.s^{-1}$ |
| $v_2$ | : Feeding speed, $mm \cdot s^{-1}$ |

| $y$ | : Position of node in y axis in global position, mm |
| $\sigma$ | : Flow stress, MPa |
| $\varepsilon$ | : Strain |
| $\dot{\varepsilon}$ | : Strain rate, s$^{-1}$ |
| $\omega$ | : Frequency, Hz |
| $\phi$ | : Phase, rad |
| $\lambda$ | : Thermal conductivity, W·mm$^{-1}$·K$^{-1}$ |
| $\sigma_e$ | : Standard deviation of error, mm |
| $\Delta v_1$ | : Changing of compression speed, mm·s$^{-1}$ |
| $\Delta v_2$ | : Changing of feeding speed, mm·s$^{-1}$ |
| $\Delta x$ | : Compression stroke |

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
