# Peer review of "A Vision-Based Fuzzy Control to Adjust Compression Speed for a Semi-Dieless Bellows-Forming"

_metals, doi:10.3390/met10060720_

Round 1

Reviewer 1 Report

This paper entitled “A Vision-based Fuzzy Control for a Semi-dieless Bellows-forming with a Local Heating Technique", the authors proposed a FEM for a semi-dieless bellows forming process with a local heating technique and, then, a vision-based fuzzy control system to improve the accuracy and repeatability of this process. In my opinion this paper is within the scope of this journal and the work present a good quality and scientific interest. Only minor revisions need to be addressed in the present form of the paper to be considered for publication in this journal.

  • The abstract does not summarize the research work and does not provide technical and numerical information about the crucial parameters, as well as the main conclusion. It is advised to improve this section.
  • The introduction section of this paper is coarse and very short. It is recommended to add novel publications to describe the state of art of this topic like the influence of this local heating techniques in metals (DOI: 10.1016/j.matdes.2015.11.067). Also, please review the following recent article and its references that have been study the thermomechanical coupling effects and the FEM model to analyze the residual stress distribution (DOI: 10.1115/1.4037798). It can also help you to strengthen the discussion section because it is a thermomechanical problem.
  • Figure 16 and 17 present some curve that are not labeled. Which is the different between the blue line and the dashed red line? Figure 18 need to mention how was estimated the error bars, is it standard deviation? Figure 20 present format error, the target of bellow height should be blue color.
  • It would be interesting to analyze the same research for pipes. How would influence the bellows height for different initial thickness of the samples?

The research work is very interesting and well documented congratulation. I hope these minor comments will help to strengthen the manuscript.

Author Response

Dear Reviewer 1

Reviewer 2 Report

  • The paper is well written and the findings of the research study proves the proposed idea clearly, however there are few spilling mistakes which I noticed in the text and also in the figures, such as line 58,62,243, Figure 7 etc. Please read out the article once more and remove the spilling mistakes.

  • The work of the paper is mainly focused on the inlet and outlet speed of the working material while the temperature is kept constant to 1100, so it is best that the authors should reconsider about the title of paper and instead of temperature using term of feeding speed will be more appropriate.

  • Although the paper is will written but there is one point which I noticed, the results of this study is mainly focused on two parameters velocity and temperature, however there is small details about the change in temperature and how its effecting the shape accuracy and height of bellow.

  • In Figure 9 there is a small difference due to two different compression values in term of bellows height, explain why?

  • The abbreviation of variables used in the Figure 12 and Table 2,3 and 4 are missing, explain more clearly, what is meant by these variables.

Author Response

Dear Reviewer 2

Reviewer 3 Report

1.The units shold be described in Eqs.(1)~(3), (6), (7).

2. In Table 1 and Fig. 5, units of heat transfer coefficent(HTC) are wrong.

3. Two HTCs (1000 vs. 40) in Fig. 4(b) and Fig. 5 are very different values.

4. Type and size of letters in Figures are not consistent.

5. In Fig.13, what kind of data do you get from image processing. And whatdata is passed to the fuzzy controller.

6. Are bellow profies from experimet compared with those from simuation?

7. Some units are wrong in Table A1.

Author Response

Dear Reviewer 3

Round 2

Reviewer 3 Report

The manuscript is precisely revised following the reviewer's comments.